# Integrated Systems Pharmacology, Urinary Metabonomics, and Quantitative Real-Time PCR Analysis to Uncover Targets and Metabolic Pathways of the You-Gui Pill in Treating Kidney-Yang Deficiency Syndrome

**DOI:** 10.3390/ijms20153655

**Published:** 2019-07-26

**Authors:** Ruiqun Chen, Jia Wang, Runhua Zhan, Lei Zhang, Xiufeng Wang

**Affiliations:** 1School of Life Sciences and Biopharmaceutics, Guangdong Pharmaceutical University, Guangzhou 510006, China; 2Shool of Pharmacy, Guangdong Pharmaceutical University, Guangzhou 510006, China; 3College of Medical Information Engineering, Guangdong Pharmaceutical University, Guangzhou 510006, China

**Keywords:** kidney-yang deficiency syndrome, urinary metabonomics, target protein, RT-qPCR, You-gui pill

## Abstract

Kidney-yang deficiency syndrome (KYDS) is a metabolic disease caused by a neuro-endocrine disorder. The You-gui pill (YGP) is a classic traditional Chinese medicine (TCM) formula for the treatment of KYDS and has been widely used to warm and recuperate KYDS clinically for hundreds of years in China. However, it is unknown whetherthe corresponding targets and metabolic pathways can also be found via using metabonomics based on one platform (e.g., ^1^H NMR) to study different biological samples of KYDS. At the same time, relevant reports on further molecular verification (e.g., RT-qPCR analysis) of these targets associated with biomarkers and metabolic pathways have not yet, to our knowledge, been seen in KYDS’s research. In the present study, a comprehensive strategy integrating systems pharmacology and ^1^H NMR-based urinary metabonomics analysis was proposed to identify the target proteins and metabolic pathways that YGP acts on KYDS. Thereafter, further validation of target proteins in kidney tissue was performed through quantitative real-time PCR analysis (RT-qPCR). Furthermore, biochemical parameters and histopathological analysis were studied. As a result, seven target proteins (L-serine dehydratase; phosphoenolpyruvate carboxykinase; spermidine synthase; tyrosyl-tRNA synthetase, glutamine synthetase; 3-hydroxyacyl-CoA dehydrogenase; glycine amidinotransferase) in YGP were discovered to play a therapeutic role in KYDS via affecting eight metabolic pathways (glycine, serine and threonine metabolism; butanoate metabolism; TCA cycle, etc.). Importantly, three target proteins (i.e., 3-hydroxyacyl-CoA dehydrogenase; glutamine synthetase; and glycine amidinotransferase) and two metabolic pathways (butanoate metabolism and dicarboxylate metabolism) related to KYDS, to our knowledge, had been newly discovered in our study. The mechanism of action mainly involved energy metabolism, oxidative stress, ammonia metabolism, amino acid metabolism, and fatty acid metabolism. In short, our study demonstrated that targets and metabolic pathways for the treatment of KYDS by YGP can be effectively found via combining with systems pharmacology and urinary metabonomics. In addition to this, common and specific targets and metabolic pathways of KYDS treated by YGP can be found effectively by integration with the analysis of different biological samples (e.g., serum, urine, feces, and tissue). It is; therefore, important that this laid the foundation for deeper mechanism research and drug-targeted therapy of KYDS in future.

## 1. Introduction

Kidney-yang deficiency syndrome (KYDS), a metabolic disease caused by a neuro-endocrine disorder, was recorded first in an early systematic and theoretical monograph existing in China, “Neijing” [1,2]. Studies have found that the pathogenesis mechanism of KYDS is mainly in the multi-level dysfunction of the hypothalamic-pituitary-target gland axis (adrenal, thyroid, and gonad). It is characterized by qi deficiency, cold limbs, decreased mobility, slow response, decreased appetite, cowered, polyuria, diarrhea, and sparse hair, etc. [3,4,5]. In contrast, the You-gui pill (YGP) is a classic traditional Chinese medicine (TCM) formula for the treatment of KYDS and has been widely used to warm and recuperate KYDS clinically for hundreds of years in China [6]. It was originally described by “Jing-Yue Complete Works” in the year 1624 and consisted of ten herbs, including *Radix Rehmanniae Praeparata* (SDH, Shu-Di-Huang), *Radix Aconiti Lateralis Prreparata* (FZ, Fu-Zi), *Cinnamomi cortex* (RG, *Rou-Gui*), *Rhizoma Dioscoreae* (SY, Shan-Yao), *Fructus Corni* (SZY, Shan-Zhu-Yu), *Semen Cuscutae* (TSZ, Tu-Si-Zi), *Fructus Lycii* (GQ, Gou-Qi), *Cervi cornus colla* (LJJ, Lu-Jiao-jiao), *Eucommiae cortex* (DZ, Du-Zhong), and *Radix Angeliccae Sinensis* (DG, Dang-Gui) [6,7,8]. The ratio of ten herbs in YGP was: SDH (8); SY (4); SZY (3); GQ (4); LJJ (4); TSZ (4); DZ (4); RG (2); FZ (2); DG (3). Modern pharmacological studies have shown that YGP has the effects of anti-inflammatory, antioxidant, immune regulation and endocrine regulation [9].

Studies have indicated that metabonomics based on different platforms (e.g., ^1^H NMR [4,5,7,10], GC/MS [2,11], LC/MS [3,12,13,14,15,16,17]) can identify some biomarkers, targets, and metabolic pathways associated with KYDS or KYDS treated by Chinese medicine. However, it is unknown whether these corresponding targets and metabolic pathways can also be found via using metabonomics based on one platform (e.g., ^1^H NMR) to study different biological samples of KYDS.At the same time, relevant reports on further molecular verification (e.g., RT-qPCR analysis) of these targets related to biomarkers and metabolic pathways have not yet, to our knowledge, been seen in KYDS’s research. Our previous studies of serum metabonomics had also discovered that YGP could play a therapeutic role by affecting certain targets and metabolic pathways of KYDS rats [7]. Based on this, we combine systems pharmacology, urinary metabonomics, and RT-qPCR analysis to study the therapeutic effects of YGP on KYDS, hoping to further discover new targets and metabolic pathways, and verify the relevant results. The results of our current study will lay the foundation for further research on the molecular mechanism of KYDS and provide reference for drug targeted therapy of KYDS.

## 2. Results

### 2.1. Biochemical Parameters and Histopathological Analysis

As shown in Figure 1A, in the KYDS group (MOD): Hypothalamic neuron cells shrunk and numbers of cells were reduced, and nucleuses were also pyknosis; pituitary congestion was obvious. Basophils also became smaller and vacuolar degeneration occurred; adrenal cortex was thinned, cells were atrophied, and cell gaps were widened; fibrous tissue in thyroid cells were proliferated, colloids in the thyroid follicular cavity were decreased, and follicular cells were shrunk, with an irregular shape; testicles were atrophied, the arrangement was loose, and the number of sperm was reduced to varying degrees; the liver developed steatosis, the liver cell nucleus shrunk, and the number decreased; the glomerulus of the kidney was atrophied or swollen, the number was reduced, the mesangium was dilated, and the tubules were dilated. These were consistent with the results of Liu et al. [12]. In contrast, the hypothalamic-pituitary-target gland axis (thyroid, adrenal gland, and testis), the liver, and kidney were improved in the YGP group to some extent after administration. Consistent with this, YGP was found to reduce exogenous glucocorticoid-induced apoptosis in anterior pituitary cells [18]. 

The weights of the rats among the three groups were changed significantly before and after administration (Appendix A). Similarly, thyroid axis [thyroid stimulating hormone (TSH), triiodothyronine (T_3_), tetraiodothyronine (T_4_)], testis [luteinizing hormone (LH), follicle stimulating hormone (FSH), testosterone (T)] and adrenal axis [adrenocorticotropic hormone (ACTH), cortisol (CORT), 17-hydorxycorticosteroids (17-OHCS)] related hormones were also found to be significantly decreased in the MOD group, but significantly increased in the YGP group (Table 1). This also indicated that the hypothalamic-pituitary-target gland axis of the KYDS model was inhibited, resulting in decreased hormone secretion and atrophy of the endocrine glands. In contrast, YGP can effectively reverse this inhibition, thereby regulating KYDS. Meanwhile, many studies have shown that 17-OHCS in 24 h urine was significantly decreased when KYDS occurs, and KYDS was clinically still determined by combining 17-OHCS and clinical symptoms [2,12,14,19]. In addition, although there was no significant difference in the superoxide dismutase (SOD) among three groups (Figure 1B), methane dicarboxylic aldehyde (MDA) increased in the MOD group, and significantly decreased in the YGP group, which partly explained that KYDS was associated with oxidative stress [20], and YGP can improve oxidative damage [21]. The level of blood urea nitrogen (BUN) and alanine aminotransferase (ALT) did not change significantly in the MOD and YGP groups, but there was a downward trend in the MOD group (Figure 1B). Aspartate aminotransferase (AST) and serum creatinine (Scr) were significantly elevated in the KYDS group and significantly decreased in the YGP (Figure 1B). Thesealso shows that KYDS is accompanied by a certain degree of liver and kidney damage, and YGP has a protective effect on the liver and kidney [22].

Resultantly, the above illustrates the successful modeling of KYDS, and KYDS is accompanied by certain liver and kidney damage. In contrast, YGP has a protective effect on KYDS and function of the liver and kidney.

### 2.2. Systems Pharmacology Analysis

Many active compounds have been widely reported to be important pharmacological components in GQ and RG, these compounds were considered as active compounds even though they owned relatively small drug-likeness (DL) and blood–brain barrier (BBB) values [23,24]. Together, 61 potentially active compounds in YGP, 3177 corresponding targets, and 201 pathways were finally screened out [7]. The interaction graph among YGP, nine kinds of TCM ingredients, and 61 kinds of active compounds is shown in Figure 2A. What is more, three targets (3-hydroxyacyl-CoA dehydrogenase, glutamine synthetase, glycine amidinotransferase) [12] and 33 pathways were discovered through literature search. Therefore, a total of 3180 targets (Figure 2B) and 234 pathways (Figure 2C) in which KYDS might be affected were included.

### 2.3. ^1^H NMR Spectra of Urine Sample

The typical ^1^H NMR NOESYPR1D spectra of urine samples among CON, MOD, and YGP groups are shown in Figure 3. The peaks of spectra were assigned according to the Human Metabolome Database (HMDB) and some metabolites obtained from literatures [25,26,27,28,29,30,31]. From the spectra, it was found that there was a significant change in the signal peak before and after YGP treatment, and 41 metabolites were ultimately assigned to these spectral signal peaks. Thus, multivariate statistical analysis was further performed on these three sets of data, and differential metabolites were searched for.

### 2.4. Multivariate Statistical Analysis

The principle component analysis (PCA) found that CON and MOD groups were significantly separated, and the YGP group was close to the MOD group (Figure 4(A1–4A2)). Thus, these three groups were subjected to orthogonal partial least squares discriminant analysis (OPLS-DA) and the important variable values (VIP > 1) in *S*-plots were sought (Figure 4(B1–4C2)). Likewise, results of permutation test and CV-ANOVA also indicate that both of modes were reliable (Appendix A). By contrast, 27 metabolites with significant changes before and after YGP treatment were finally discovered (Figure 5, Appendix A). It can be found from Figure 5 that these metabolites were mainly related to amino acid metabolism (18.5%), energy metabolism (37%), ketone body metabolism (7.4%), fatty acid metabolism (14.8%), methane metabolism (18.5%), and glycoprotein (3.8%).

### 2.5. Metabolic Pathway Analysis

These 27 significantly altered metabolites were introduced into MetaboAnalyst 4.0 for pathway analysis. The results of analysis revealed that these metabolites were mainly associated with 29 metabolic pathways (Appendix A). Among them, there were eight metabolic pathways with *p* < 0.05 (Figure 6A). These eight pathways were considered to be key metabolic pathways that YGP affected KYDS.

Subsequently, eight key metabolic pathways were then mapped to pathways discovered by systems pharmacology analysis. Then, target proteins and active compounds corresponding to YGP from systems pharmacology were reversely searched from the overlapping pathways, and finally the potential active compounds, target proteins, and metabolic pathways of YGP affecting KYDS were got. It was found that there were seven metabolic pathways overlapping with it (except methane metabolism), and the corresponding five TCM ingredients, twelve active compounds, and eight targets could be acquired by reverse mapping analysis (Figure 6B,C). The network of YGP–herbs–active compounds–target proteins–pathways can be seen in Figure 6C. From this network, we can obviously find the relationship between the genes and pathways. 

By comparing the results of this study (urine sample) with previous studies (serum sample) metabonomics and systems pharmacology, we could find between the two that five Chinese herbal ingredients, twelve active compounds, five targets, and three metabolic pathways were the same (Figure 6D). Therefore, it could be preliminarily stated that YGP mainly acted on ten metabolic pathways through five TCM ingredients, eighteen active compounds, and eleven targets, so as to play a role in improving and regulating KYDS. Taken together, in combination with three target proteins (spermidine synthase, alcohol dehydrogenase 1C, and alcohol dehydrogenase class-3) found in previous studies [7], we performed RT-qPCR analysis on eleven target proteins (Table 2) to verify genes with significant differences or changing trends. 

### 2.6. YGP Regulates KYDS-Related Differential Gene Expression

Results of RT-qPCR analysis revealed that six genes (i.e., GTP, Yars, Srm, Hadh, Glul, and Sds) showed a significant decrease and one showed a decrease trend (i.e., Gatm) in the MOD group compared with the CON group (Figure 7). Compared with the MOD group; however, one gene (Srm) in the YGP group increased significantly, and five genes (GTP, Yars, Hadh, Gatm, and Sds) showed an upward trend. Besides, two kinds of genes were not expressed among three groups, including BHSM and CBS. As seen, Figure 7 only shows one gene was statistically different between the MOD and YGP groups. However, the other five genes had an increasing tendency in the YGP group, and their *p*-value was also close to 0.05. Therefore, we believe that these targets should also be significantly different in the case of expanding the sample size. 

### 2.7. Correlation Analysis between Differential Metabolites and Biochemical Parameters

Figure 8 shows the results of correlation analysis of 27 differential metabolites and nine biochemical parameters related to the pathogenesis axis of KYDS. Those correlation coefficient values (|*r*| ≥ 0.70, *p* < 0.05) were considered to be significantly correlated. It could be seen from the figure that there was a difference in the correlation between metabolites and biochemical indicators before and after administration. From the perspective of biochemical indicators, the number of biochemical indicators and metabolites in the CON group (six: TSH, T_4_, LH, FSH, T, CORT) and the YGP group (seven: T_3_, LH, FSH, T, ACTH, CORT, 17-OHCS) were significantly more relevant than the MOD group (five: T_3_, T_4_, T, ACTH, 17-OHCS). From the perspective of metabolites, there was more significant correlation of biochemical indicators and metabolites between the CON group and the YGP group, while the MOD group was relatively single. In short, when KYDS occurs, the complex correlation between metabolites and biochemical indicators was broken and reduced, leading to extreme metabolic disorders (i.e., some relations between biochemical indicators and metabolites became absolutely positive correlation (T, ACTH) or negative correlation (T_4_, 17-OHCS), and some were completely irrelevant (TSH, LH, FSH, and CORT)). After administration of YGP; however, the correlation between biochemical indicators and metabolites increased, and it tended to the original homeostasis in the CON group.

### 2.8. Overall Interactive Network Diagram of Herbs, Active Compounds, Targets, Pathways, and Metabolites

Figure 9 shows the overall interactive network diagram of herbs, active compounds, targets, pathways, and metabolites. Notably, it could be clearly found that what targets, metabolic pathways, and metabolites were affected by the active components of YGP. Combining systems pharmacology, urinary metabonomics, and RT-qPCR techniques to analyze the urine sample of YGP-treated KYDS rats, we found that YGP mostly affects thirteen metabolites through five herbs, ten active compounds, seven targets, and eight metabolic pathways, thereby playing a role in treating KYDS (Figure 9). The mechanism of action primarily involved energy metabolism, oxidative stress, ammonia metabolism, amino acid metabolism, and fatty acid metabolism. The 2D structure diagram of ten active compounds in YGP is displayed in Appendix A. 

## 3. Discussion

### 3.1. Energy Metabolism

The body provides energy largely through glycolysis and gluconeogenesis. Studies have shown that under the interference of high doses of hydrocortisone, the hypothalamic monoamine transmitter in rats is activated and promotes the “over-consumption” of energy and immune system, which ultimately leads to a state of “exhaustion” similar to KYDS in TCM [5,11]. In addition, our previous study also found that the “exhaustion” of KYDS is mainly caused by damage to the mitochondria and liver of rats, thus affecting the body’s energy metabolism (TCA cycle and gluconeogenesis) [7,10]. On the contrary, a study has demonstrated that YGP exhibits anti-apoptosis effects (induced by exogenous glucocorticoid) via regulating mitochondrial function in apoptosis pathways of anterior pituitary cells [18]. Li et al. also find that there is an imbalance between aerobic metabolism and anaerobic glycolysis in patients with Yang-deficiency, resulting in a disorder of energy metabolism, which in turn leads to the downregulation of creatinine, lactate, and citrate [32]. This is consistent with the variation trend in creatinine, lactate, and citrate in the MOD group of this study.

L-serine dehydratase (Sds) catalyzes the deamination of L-serine to yield pyruvate, and is involved in the gluconeogenesis pathway, which is part of carbohydrate biosynthesis [33]. In MOD group, the level of L-serine dehydratase decreased significantly, while it increased in YGP group. 24-Methylcholest-5-enyl-3belta-O-glucopyranoside_qt in SY, 11, 14-eicosadienoic acid in FZ and (E, E)-1-ethyl octadeca-3, 13-dienoate in GQ can act synergistically on L-serine dehydratase to affect valine, leucine and isoleucine biosynthesis, and glycine, serine and threonine metabolism. Among which, glycine, serine and threonine metabolism has been found to be one of the main perturbed pathways in the pathological process of KYDS, JinkuiShenqi Pill; however, can play a therapeutic role in KYDS by improving this metabolic disorder [15]. Therefore, the levels of valine and glycine decreased in the MOD group, and increased in the YGP group.

Phosphoenolpyruvate carboxykinase (GTP) catalyzes the conversion of oxaloacetate (OAA) to phosphoenolpyruvate, and this is the rate-limiting step in the gluconeogenic pathway [7]. Studies have found that KYDS, in addition to the inhibition of the neuroendocrine system, was accompanied by a certain degree of damage to the liver and kidney [5,34]. Lu et al. also have shown that warm and tonify kidney-yang herbs have protective effects on the liver mitochondrial proteome of kidney-yang deficiency rats [35]. Meanwhile, the experiment shows that YGP has no liver and kidney toxicity, and can protect the injured liver and kidney [22]. Ethyl oleate (NF) and mandenol in SZY, atropine and mandenol in GQ, kadsurenone and denudatin B in SY can affect the TCA cycle by acting on phosphoenolpyruvate carboxykinase. Clinical studies have also shown that KYDS is associated with inhibition of the TCA cycle [17]. What is more, non-glycemic substances (e.g., valine, alanine, lactate, and glycerol) can be converted to pyruvate by gluconeogenesis, and eventually yield glucose to maintain a constant blood glucose level. Studies have also shown that when KYDS occurs, glucocorticoid secretion is reduced, thereby inhibiting gluconeogenesis and accelerating glucose catabolism, resulting in lower blood glucose (decreased levels of α-glucose and β-glucose) [36,37]. This also shows that YGP can enhance the TCA cycle and gluconeogenesis by improving damage of the mitochondria and liver, thus ensuring the normal metabolism of energy.

Glycine amidinotransferase (Gatm) in mitochondrial catalyzes the biosynthesis of guanidinoacetate which is the immediate precursor of creatine [38]. Creatine plays a vital role in energy metabolism in muscle tissues [39]. Both creatine and creatinine showed a significant decrease in MOD group and increased in YGP group. Betaine in GQ can affect glycine, serine, and threonine metabolism by acting on glycine amidinotransferase. Meanwhile, betaine has been shown to be the main bioactive element of *Fructus Lycii* (GQ) [40]. This also indicates that energy metabolism is inhibited when KYDS occurs, while YGP can improve energy metabolism. What is more, creatine plays a role in central nervous system (CNS) development [41], and KYDS is closely related to neuroendocrine disorders. Studies have also shown that YGP exerted the potent neuroprotective effects on the CNS [42,43,44]. This manifests that the CNS is inhibited when KYDS occurs, YGP; however, can regulate and reinforce the development of the CNS by affecting the production of creatine and glycine amidinotransferase. Phenylalanine, a metabolite used in the synthesis of important neurotransmitters and hormones, such as dopamine, epinephrine, and norepinephrine, showed a significant decrease in the MOD group and a significant increase in the YGP group, which also confirmed this conclusion [45].

### 3.2. Oxidative Stress

Spermidine synthase (Srm) is an enzyme thatcatalyzes the transferof the propylamine group from S-adenosylmethioninamine to putrescine in the biosynthesis of spermidine. Spermine and spermidine are polyamine that mainly exists in human semen that mediates protection against oxidative damage caused by hydrogen peroxide [46]. Spermidine synthase decreased significantly in the MOD group and significantly increased in the YGP group. Clinical study has demonstrated that the administration of YGP decoction can enhance male sperm fertilizing ability, which was achieved by increasing sperm acrosin activity [47]. Wu et al. also shows the loss of spermidine synthase reduces the growth rate and renders the cells more sensitive to oxidative damage [48]. Simultaneously, it has been reported that hydrocortisone can increase MDA levels, reduce SOD, and lower molecular scavenger glutathione (GSH) levels, indicating that hydrocortisone can induce oligospermia by strengthening internal oxidative stress [49,50]. This is consistent with the results of biochemical analysis, and pathological analysis of testicular tissue in the previous part of this study (Figure 1). In addition, studies have also found that lipid peroxidation damage is the pathophysiological basis of kidney-yang deficiency syndrome [51]. The significant increase of plasma lysophosphatidylcholine (LPC) levels in model rats implied that “kidney-yang deficiency syndrome” involved the oxidant–antioxidant balance in vivo, and it was also reported that icariin significantly inhibited the release of reactive oxygen species (ROS) to reduce oxidative stress [13]. Correspondingly, YGP can also improve the lipid peroxidation stress response ability of hypothyroidism model rats, and have a certain effect on alleviating free radical peroxidation damage in vivo [22]. Atropine in GQ can affect arginine and proline metabolism by acting on spermidine synthase. Hence, insofar as I can judge, YGP can reduce the oxidative stress damage of rats’ DNA by activating spermidine synthase and promoting the production of spermine and spermidine.

### 3.3. Ammonia Metabolism

Glutamine synthetase (Glul) is an enzyme that plays an essential role in the metabolism of nitrogen by catalyzing the condensation of glutamate and ammonia to form glutamine [52]. It is present predominantly in the brain, kidney, and liver. Glutamine synthetase significantly decreased in the MOD group, indicating that the metabolism of nitrogen was inhibited. As a result, a significant decrease in nitrogen-containing compounds (i.e., dimethylamine, NAG, TMAO) and amino acids occurred. Betaine in GQ can affect alanine, aspartate, and glutamate metabolism, and glyoxylate and dicarboxylate metabolism by acting on glutamine synthetase. Liu et al. have also confirmed that betaine can act as an active compound on glutamine synthetase to play a role in the treatment of KYDS [12]. Furthermore, ammonia and glutamate produced by various tissues in vivo can be converted to glutamine under the action of glutamine synthetase. This is an important way to detoxify ammonia and is also a form of transport and storage of ammonia. Zou et al. also shows that the body’s ammonia transport function was blocked under the state of KYDS [5]. Additionally, glutamine synthetase can be activated by α-ketoglutarate. By contrast, α-ketoglutarate decreased in the MOD group, whereas both glutamine synthetase and α-ketoglutarate increased significantly in the YGP group. Currently, our previous studies also found that the levels of glutamate and glutamine decreased in the presence of KYDS, but increased in the YGP group [7]. This illustrates that glutamine synthetase is inhibited, causing the ammonia metabolism to be blocked, which in turn leads to the damage of CNS [7,19]. Likewise, a wealth of studies have shown that there are a number of links between changes in glutamine synthetase activity and neurodegenerative disorders, such as Alzheimer’s disease [53,54]. Consistent with this, ammonia is synthesized in the liver and is excreted in the form of ammonium in the kidney. When KYDS occurs, the liver and kidney are damaged, which causes the metabolism of ammonia to be blocked [5]. In contrast, YGP can protect the liver, kidney, and CNS, thus ensuring the normal metabolism of ammonia in vivo [6].

### 3.4. Amino Acid Metabolism

Tyrosyl-tRNA synthetase (Yars) recognizes specific amino acids and plays an important role in the amino acid synthesis of proteins. It developed a significant decrease in the MOD group and increased in the YGP group. A myriad of studies have also shown that when KYDS occurs, amino acid metabolism is disrupted [5,7,10,13]. This indicates that various proteins and enzymes required for the synthesis of the body are inhibited, resulting in abnormal metabolism of life. However, styrone in RG can affect aminoacyl-tRNA biosynthesis by increasing the activity of tyrosyl-tRNA synthetase, thereby promoting protein production. Clinical studies have shown that the level of NAG in the blood of patients with Yang-deficiency is lower than that of normal people, indicating a reduction in the synthesis of this glycoprotein [32]. NAG was found to be significantly decreased in the MOD group, and a significant increase in the YGP group also verified this result. Moreover, studies have shown that oxidative stress will induce tyrosyl-tRNA synthetase to rapidly translocate from the cytosol to the nucleus to protect against DNA damage [55]. To some extent, this also explains the occurrence of oxidative stress in KYDS rats. Meanwhile, studies have also shown that styrone is one of the main components of RG volatile oil, which has anti-oxidation and anti-tumor effects [7,56]. These again manifest that YGP can reduce oxidative stress damage by increasing the production of tyrosyl-tRNA synthetase.

### 3.5. Fatty Acid Metabolism

3-hydroxyacyl-CoA dehydrogenase (Hadh) plays an essential role in the mitochondrial β-oxidation of short chain fatty acids, which is part of lipid metabolism [57]. β-oxidation of mitochondrial fatty acids was the energy source of multiple organs and tissues, whose cytomembranes were composed of phosphatide, including two fatty acids [58]. Remarkably, short chain fatty acids (2-hydroxy isovalerate, 2-oxoisocaproate, acetate and formate) decreased in the MOD group, while they increased in the YGP group. Besides, 3-hydroxyacyl-CoA dehydrogenase significantly decreased in the MOD group and increased in the YGP group. This indicates that when KYDS occurs, β-oxidation of short chain fatty acids in the mitochondrial is inhibited, and energy metabolism is disordered [17,35]. Liu et al. have shown that mitochondrial injury and fatty acid metabolism abnormalities in KYDS rats will finally result in a decrease in the level of fatty acids [12]. Huang et al. [13] also have proved that the lipid metabolism is disordered when KYDS occurs. This is consistent with the results of clinical studies [32]. Conversely, denudatin B in SY can affect butanoate metabolism by improving the activity of 3-hydroxyacyl-CoA dehydrogenase. This effect may be due to the fact that YGP improves the mitochondrial function of KYDS, thereby increasing the activity of 3-hydroxyacyl-CoA dehydrogenase. The biological derivative compound of fatty acids (e.g., acetylcarnitine) may be closely related to KYDS, because it possesses a neuroprotective role on ischemic brain damage and can improve the neurological symptoms, reduce the free radical-mediated protein oxidation, and restore the brain energy metabolites [12,59,60]. Tryptophan, an essential amino acid of protein biosynthesis, is primarily involved in two metabolic pathways: The kynurenine pathways and serotonin (5-HT) formation, which affects the function of neurotransmitters, neural circuits, and neuromodulator in the body [19,45]. It was evident that tryptophan decreased in the MOD group and increased in the YGP group. Coherently, this again illustrates the damage to the nervous system in the KYDS rats, while YGP has a protective effect on the CNS [42,43].

Taken together, compared with the serum metabonomics and systems pharmacological analysis of YGP in the treatment of KYDS [7], five herbs and ten active compounds acting on KYDS in this study are the same, but the active compounds are less than found in previous studies. Besides, we find three new target proteins (i.e., 3-hydroxyacyl-CoA dehydrogenase; glutamine synthetase; and glycine amidinotransferase), and three target proteins (i.e., phosphoolpyruvate carboxykinase; tyrosyl-tRNA synthetase; and L-serine dehydratase) are simultaneously found in urine and serum samples. The spermidine synthase; however, is discovered in serum samples of KYDS rats uniquely. What is more, by comparing the seven targets and eight pathways identified in this study with the metabolomics results of KYDS rats (serum, urine) [2,3,4,5,7,10,11,12,13,14,15,19] and clinical patients (seminal plasma, serum, and urine) [17,32] through academic search, we find that six metabolic pathways are consistent with the results of previous research, and two metabolic pathways (butanoate metabolism; glyoxylate and dicarboxylate metabolism), to our knowledge, had been newly discovered in this study. The results of serum and urine metabonomics combined with systems pharmacology showed that only five ingredients of YGP (i.e., SZY, GQ, SY, FZ and RG) directly regulate KYDS, while the other five ingredients (i.e., SDH, DZ, LJJ, TSZ, and DG) did not participate in the adjustment. This may be due to the limitations of the methods and techniques used in this study. These five TCM ingredients that are not involved in KYDS regulation are commonly-used nourishing herbs, and have a variety of effective components, which can regulate the body’s nervous system, endocrine system, immune system, liver and kidney function, and oxidative stress, etc. Therefore, we believe that the other five TCM ingredients in YGP should also play a certain (indirect or co-regulation) role in the treatment of KYDS, but the specific mechanism of action remains to be further studied. Further research on this indirect or co-regulatory role is also a potential direction for research and development of drugs in the future.

Additionally, although methane metabolism is not in the overlapping metabolic pathway, it should also play a role in the development of KYDS. Methane metabolism is mainly related to the methyl transfer reaction of dimethylamine (DMA), trimethylamine (TMA), trimethylamine-*N*-oxide (TMAO), and betaine. Zou et al. also have reported that the transmethylation reactions in KYDS rats is inhibited, resulting in a decrease in betaine levels [5,37]. In the present study, betaine, DMA, and TMAO also decreased significantly in the MOD group, but increased significantly in the YGP group. This all indicates that KYDS is related to the disorder of methane metabolism, and YGP can restore methane metabolism. Clearly, further research is needed to shed light on the role of methane metabolism in the development of KYDS, particularly the transmethylation reactions. 

Given some of the herbs, active compounds, targets, metabolic pathways, and differential metabolites that YGP acts on KYDS are found in this study by integrating systems pharmacology, urinary metabonomics, and RT-qPCR analysis, more subsequent experiments and assays, such as samples of clinical patients, western blot, targeted metabonomics, and molecular pharmacology, are needed to verify the relevant results from different levels. Besides, these metabolites (biomarkers) identified by metabonomics directly reflect the status of related upstream proteins. So, it is interesting to construct metabolite-protein interaction and validate the expression of the valuable proteins, besides of the proteins (targets) predicted by system pharmacology [61,62]. Last but not least, metabonomics studies on the liver and kidney tissues will also contribute to a more comprehensive and in-depth understanding of the occurrence of KYDS and the regulatory mechanism of YGP on KYDS.

## 4. Materials and Methods

### 4.1. Chemicals and Reagents

Hydrocortisone injections (0.5 mL/100 g) were bought from Tianjin Jinyao Pharmaceutics Co., Ltd. (batch number: 1801161, Tianjin, China). Ten kinds of Chinese medicine ingredients of YGP were from Baihetang Pharmaceutical Co., Ltd. (Guangzhou, China), and authenticated by Associate Professor Hongyan Ma (College of Traditional Chinese Medicine of Guangdong Pharmaceutical University). Batch number corresponding to all Chinese medicine ingredients: SDH (160501), SY (170101), SZY (160401), GQ (20171114), LJJ (170603), TSZ (170101), DZ (170401), DG (170401), RG (170601), FZ (170308). Real-time PCR LightCycler instrument was provided by Roche Diagnostics Co., LTD (Basle, Switzerland). Reverse transcription kits, and SYBR Premix Ex Taq™ kit were obtained from Takara Biotechnology Co., LTD (Tokyo, Japan). RNase-free water was provided by Ambion (Austin, TX, USA). Five-millimeter nuclear magnetic tube and deuterium oxide (D_2_O, including 0.05% TSP) was purchased from Qingdao Tenglong Technology Co., LTD (Qingdao, China). All primers used were synthesized and supplied by Sangon Biotech. Co. LTD (Shanghai, China). A 500 MHz Bruker AVANCE III NMR instrument (Bruker, Swiss) was provided by Center Laboratory of Guangdong Pharmaceutical University. Distilled water (18.2 MΩ) was from a Milli-Q water purification system (Millipore, MA, USA). Biochemical indicators included thyroid stimulating hormone (TSH), triiodothyronine (T_3_), tetraiodothyronine (T_4_), luteinizing hormone (LH), follicle stimulating hormone (FSH), testosterone (T), adrenocorticotropic hormone (ACTH), and cortisol (CORT) were obtained by entrusting the BeituDongya Biotechnology Research Institute (Beijing, China). 17-hydorxycorticosteroids (17-OHCS), methane dicarboxylic aldehyde (MDA), superoxide dismutase (SOD), aspartate aminotransferase (AST), alanine aminotransferase (ALT), serum creatinine (Scr) and blood urea nitrogen (BUN) were detected via entrusting the Nanjing Jiancheng Bioengineering Institute (Nanjing, China). 

### 4.2. Systems Pharmacology

The methods to screen potential active compounds and corresponding target proteins in YGP via systems pharmacology were according to our previous study [7]. Briefly, nine Chinese medicines (except LJJ) in YGP were introduced into the Traditional Chinese Medicine Systems Pharmacology Database and Analysis Platform (TCMSP) to screen the corresponding potential active compounds. The screening criteria included oral bioavailability (OB) ≥ 30%, blood–brain barrier (BBB) value > 0.3, drug-likeness (DL) ≥ 0.18, and a half-life (HL) ≥ 4. Among these parameters, OB represented the percentage of an orally administered dose of unchanged drug that reached the systemic circulation, which revealed the convergence of the ADME (absorption, distribution, metabolism, and excretion) process [63]. The function of BBB was to limit the passage of protein and potentially diagnostic and therapeutic agents into the brain parenchyma [64], and compounds with BBB > 0.3 were considered as strong penetrating (BBB+). DL was used to estimate how “drug-like” a prospective compound was, which helped to optimize pharmacokinetic and pharmaceutical properties [65]. The “drug-like” level of the compounds was 0.18, which was commonly used as a selection criterion for the “drug-like” compounds in the traditional Chinese herbs. HL was arguably the most important property as it dictated for the timescale over which the compound might elicit therapeutic effects [66,67]. The compounds with HL ≥ 4 were regarded as the mid-elimination group. All these given screening criteria were based on the TCMSP database’s suggested drug screening criteria (http://lsp.nwu.edu.cn/load_intro.php?id=29). After that, all the potential active compounds were introduced into the PharmMapper database to predict the corresponding targets. The screening criteria for potential targets was *Z*-score ≥ 0.8 [68]. Ultimately, these potential target proteins were further introduced into the KEGG database (https://www.kegg.jp/) to annotate the KEGG pathways.

### 4.3. Preparation of YGP Solution

The preparation procedure of YGP was shown as follows [7,18]: Nine kinds of herbs, including SDH (24 g), FZ (6 g), RG (6 g), SY (12 g), SZY (9 g), TSZ (12 g), GQ (12 g), DZ (12 g), and DG (9 g) were soaked for 0.5 h in distilled water and boiled for 20 min, followed by simmering for 20 min, and then filtration. After that, the residual herbs were soaked in cold distilled water for 1 h and subjected to the same decoction and filtration procedure. On this basis, herb LJJ (12 g) was added to the mixing decoction, and further concentrated to 111 mL in a rotary evaporator. Lastly, YGP extract solution containing 1.0 g mL^−1^ of crude herb was obtained and the administration dosage for rats was 1.0 mL/100 g. The preparation procedure for YGP was according to the original composition and preparation method of YGP recorded in “Jing-Yue Complete Works” at Ming dynasty and some published references [6,69].

### 4.4. Animal Care and Experiments

Twenty-four male Sprague-Dawley (SD) rats (SPF grade, weighing 180 ± 20 g) were purchased from Experimental Animal Center of Guangzhou University of Chinese Medicine (License: SCXK (Yue) 2013-0034), and kept in SPF-grade experimental animal houses with standard animal conditions of temperature (24 ± 2 °C) and relative humidity (55 ± 5%), and a 12 h light–dark cycle in Experimental Animals Center of Guangdong Pharmaceutical University. Animals had free access to regular food and water, and were acclimatized to the new environment for seven days prior to experimentation. Afterwards, all rats were divided into three groups (*n* = 8) according to a random number table method, including the control group (CON), KYDS group (MOD), and YGP group (YGP). The specific modeling and administration methods were as follows [7]: Rats of the control group were given an intramuscular injection of 0.3 mL of 0.9% saline into a hind limb for 22 days in succession. A total of 0.5 mL/100 g hydrocortisone was injected intramuscularly into a hind limb of the MOD group rats for 22 days in succession. Likewise, the YGP group was modeled in the same manner as the MOD group, but YGP solution was simultaneously administered from the eighth day of modeling. The administration dosage was 1.0 mL/100 g for 15 days in succession. All injections were taken once a day and were carried out in the morning, and all administration also were once a day and were performed in the afternoon. The concentration of hydrocortisone was 5 g/L. All animal treatments and experiments were approved by the Animal Ethics Committee of Guangdong Pharmaceutical University, and were strictly performed in accordance with the National Institutes of Health Guide for the Care and Use of Laboratory Animals. 

### 4.5. Sample Collections

At the end of the twenty-third day, 24 h urines of all rats were collected in metabolic cages (one rat in per cage). Meanwhile, all rats were fasted but had free access to water for 12 h before urine samples were collected. The urine samples were centrifuged at 4000 rpm for 15 min, and transferred to a 2 mL centrifuge tube. Then, 10 μL of 1% sodium azide (NaN_3_) was added per EP tube and frozen in a −80 °C refrigerator. Meanwhile, the variation of weights of the rats among the three groups was also recorded and analyzed.

After collecting, all rats were sacrificed in parallel and blood was taken from the femoral artery. The upper serum was taken and stored at −80 °C refrigerator after centrifugation (4500 rpm for 10 min, 4 °C) for urinary metabonomics and biochemical analysis. Biochemical indicators included TSH, T_3_, T_4_, LH, FSH, T, ACTH, CORT, 17-OHCS, MDA, SOD, ALT, Scr, and BUN. At the same time, the hypothalamus, pituitary, thyroid, adrenal gland, testis, liver, and kidney of all rats were collected for histopathological analysis. Briefly, the hypothalamus, pituitary, thyroid, and adrenal gland were fixed overnight with 10% formalin solution, while testicular tissue was fixed overnight with Bouin’s fluid. Thereafter, it was embedded in paraffin wax and cut into 3 mm tissue sections with a microtome. Finally, all tissue sections were stained with hematoxylin and eosin (H&E), and photographed with an optical microscope.

### 4.6. Urinary Metabonomics

Urine samples were thawed at room temperature and centrifuged at 4 °C for 10 min at 5000 rpm, and then subjected to ^1^H NMR analysis. A total of 300 μL supernatant and 100 μL 0.2 mol/L phosphate buffer solution (Na_2_HPO_4_/NaH_2_PO_4_, pH = 7.4) were added into an EP tube. Meanwhile, 150 μL of D_2_O (including 0.05% TSP) was added for the deuterium lock. After being thoroughly mixed, the mixture was centrifuged (14,500 rpm, 5 min, 4 °C). 500 μL of supernatant was pipetted into a 5 mm NMR tube and stored in a 4 °C refrigerator for further analysis.

The ^1^H NMR spectra of all urine samples were recorded on a Bruker spectrometer (Avance III 500 MHz) at 298 K. A one-dimensional water pre-saturated standard NOESYPR1D pulse sequence (recycle delay-90-*t*_1_-90-*t*_m_-90-acquisition) was used in order to obtain signals with low molecular weight selectively. A total number of 128 transients were collected into 32 k data points using a spectral width of 10 kHz with a relaxation delay of 3 s and the total echo time (2*nτ*) of 100 ms. An exponential function corresponding to a line broadening factor of 0.3 Hz was applied to all acquired free induction decays (FIDs) before Fourier transformation. Usually, Carr–Purcell–Meiboom–Gill (CPMG) pulse sequence was mainly used to selectively inhibit the macromolecular signal (e.g., nucleic acid and protein) in serum or plasma samples, so as to obtain the signal peak of small molecules, while the pre-saturated NOESYPR1D pulse sequence could specifically inhibit the water peak of urine, feces, and tissue extracts, thus reducing the influence of water peak.

All of the ^1^H NMR spectra of samples were manually phased and baselined using the TopSpin software (Version 2.1, Bruker Biospin, Ettlingen, Germany), and automatically integrated by using AMIX software (V3.7.3, Bruker Biospin, Ettlingen, Germany). The integral interval was *δ* 0.5–9.0 and the integral spacing was 0.004 ppm. Besides, in order to eliminate the influence caused by the water peak, the integral value of *δ* 4.7–5.2 were set to zero. Similarly, the integral values were normalized in order to eliminate the analysis error caused by the difference in the concentration of samples. The normalized values were further introduced as raw data into the software SIMCA-P 13.0 (Umetrics, Umeå, Sweden) for the principle component analysis (PCA) and orthogonal partial least squares discriminant analysis (OPLS-DA).

The PCA analysis can be used to initially determine the clustering of the overall samples and thereby distinguish whether there are differences among different groups. Besides, the PLS-DA model was validated with a permutation test (200 permutations), and OPLS-DA was validated by both a seven-fold cross-validation and ANOVA of the cross-validated residuals (CV-ANOVA) methods [70,71]. Simultaneously, differential metabolites were extracted from *S*-plots based on their contribution to the difference. Together, variables with VIP > 1 and *p* < 0.05 in OPLS-DA analysis were screened as differential metabolites. Endogenous metabolites of urine were assigned according to the HMDB database (http://www.hmdb.ca/) and some published literature [25,26,27,28,29,30,31].

Metabolic pathways of differential metabolites were performed with MetaboAnalyst 4.0 (http://www.metaboanalyst.ca/). Metabolomics pathway analysis (MetPA) combines several advanced pathway analysis programs, such as KEGG, SMP, etc., to perform topological characterization analysis of metabolic pathways. Therefore, metabolic pathways with the strongest correlation with the metabolite group can be determined more accurately. The specify pathway analysis algorithms: Hypergeometric test was used for over representation analysis, and relative-betweenness centrality was used for pathway topology analysis. Besides, metabolic pathways with *p* < 0.05 were considered to be pathways that significantly affected the development of KYDS, and were further analyzed [7,70].

### 4.7. Integrated Systems Pharmacology and Urinary Metabonomics Analysis

The metabolic pathways of *p* < 0.05 obtained from the urinary metabonomics analysis were mapped to pathways acquired by the systems pharmacology to uncover pathways in which the two were coincident. Then, target proteins and active compounds corresponding to YGP from systems pharmacology were reversely searched from the overlapping pathways, and finally the potential active compounds, target proteins, and metabolic pathways of YGP affecting KYDS were got. On this basis, these potential targets were further verified by RT-qPCR, and differential target proteins were found. In addition, three target proteins (spermidine synthase, alcohol dehydrogenase 1C, and alcohol dehydrogenase class-3) that have been shown to affect KYDS by YGP in our previous studies were also verified by RT-qPCR in this study [7].

### 4.8. *RNA Extraction and* RT-qPCR Analysis

Total RNA was extracted from flash-frozen kidney tissues using TRIzol reagent (Invitrogen, Carlsbad, CA, USA) according to the manufacturer’s instructions. Then, RNA was reversely transcribed to cDNA by using reverse transcription kits (Takara). RT-qPCR and data collection were performed with PCR Master Mix (Takara) on Applied Biosystems (ABI) 7500. The specific PCR amplification was carried out with a 3 min pre-denaturation at 94 °C and 40 cycles (94 °C for 5 s, 64 °C for 20 s, and 72 °C for 30 s), followed by a 2 min extension at 72 °C, and 5 min cool down at 10 °C. Target mRNA expression in each sample was normalized to the housekeeping gene (β-actin) to normalize the starting cDNA levels. The 2^−ΔΔC*t*^ method was used to calculate relative mRNA expression levels. The PCR primers were listed in Table 2.

### 4.9. Pearson’s Correlation Analysis of 27 Metabolites and Nine Biochemical Indicators

To further analyze the relationship between these differential metabolites and biochemical indicators, we performed Pearson’s correlation analysis on 27 differential metabolites and nine biochemical indicators related to the pathogenesis axis of KYDS among three groups. Those correlation coefficient values (|*r*| ≥ 0.70, *p* < 0.05) were considered to be significantly correlated, and the significant change was expressed as * *p* < 0.05, ** *p* < 0.01 (two-sided test). Red indicated positive correlation, blue represented a negative correlation. Furthermore, all differential metabolites were clustered analysis by Euclidean distance.

### 4.10. Statistical Analysis

The biochemical and metabonomic data were screened by SPSS 17.0 (IBM, Armonk, NY, USA) and GraphPad Prism 5.0 (GraphPad Software, Inc., San Diego, CA, USA) software with Student’s *t* test or one-way analysis of variance (ANOVA) and Bonferroni multiple comparisons. *p* < 0.05 was considered as statistically significant differences.

## Figures and Tables

**Figure 1 ijms-20-03655-f001:**
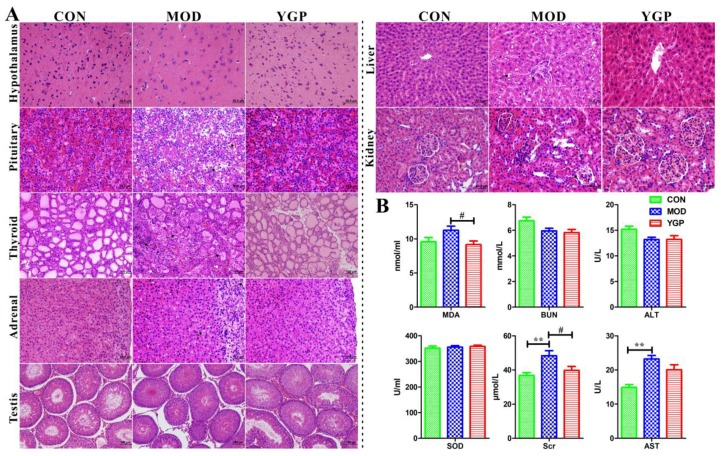
Typical histopathological changes of hypothalamic-pituitary-target gland axis (**A**), and the changes of biochemical indexes related to oxidative stress, and function of the liver and kidney (**B**). The magnification of hypothalamus, pituitary, adrenal gland, liver and kidney is 40×, while the magnification of thyroid and testis is 20×. * As compared with the control group (CON), * *p* < 0.05, ** *p* < 0.01, ^#^ as compared with the KYDS group (MOD), ^#^
*p* < 0.05, ^##^
*p* < 0.01.

**Figure 2 ijms-20-03655-f002:**
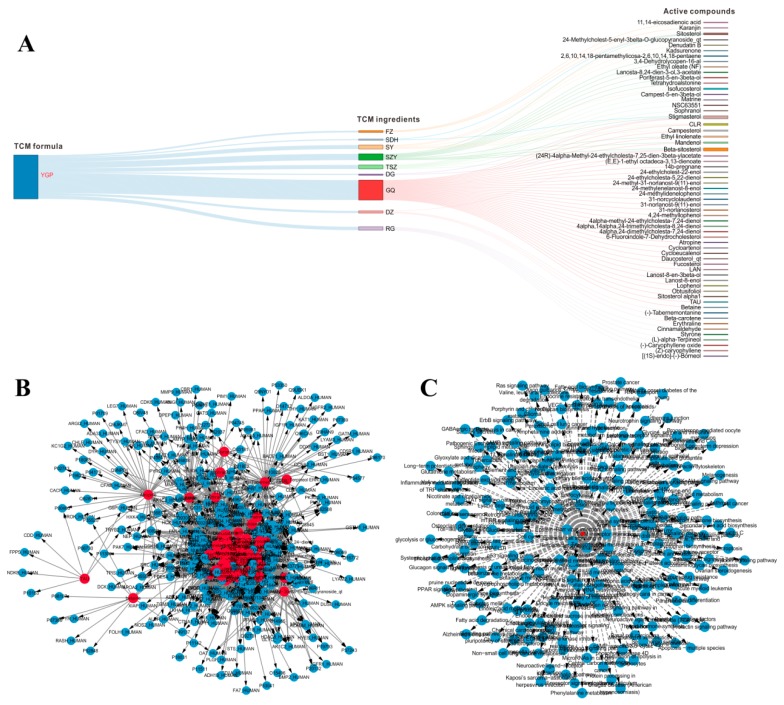
(**A**) The interaction graph among YGP, nine kinds of TCM ingredients, and 61 kinds of active compounds. (**B**) Sixty-one active compounds (red) and 3180 targets (cyan) network. (**C**) Sixty-one active compounds (red) and 234 pathways network (cyan).

**Figure 3 ijms-20-03655-f003:**
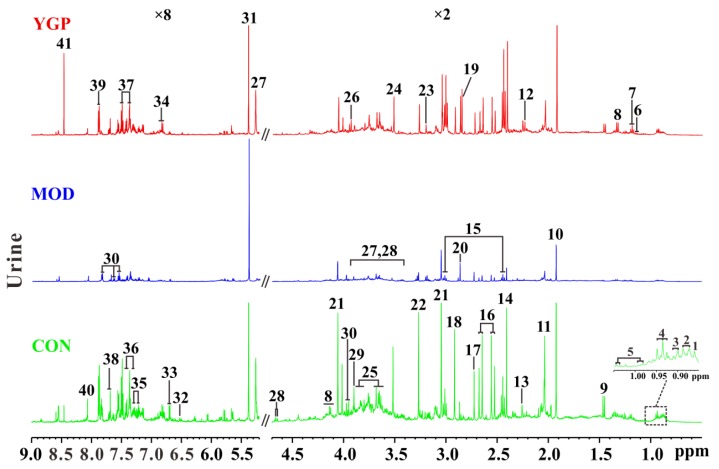
Typical ^1^H NMR spectra of urine sample (CON: Control group; MOD: KYDS group; YGP: You-gui pill group). (1) 2-hydroxyisovalerate; (2) 3-methyl-2-ketovalerate; (3) 2-hydroxyvalerate; (4) 2-oxoisocaproate; (5) valine; (6) isobutyrate; (7) 3-hydroxybutyrate; (8) lactate; (9) alanine; (10) acetate; (11) *N*-acetyl glycoprotein; (12) acetone; (13) acetoacetate; (14) succinate; (15) α-ketoglutarate; (16) citrate; (17) dimethylamine (DMA); (18) dimethylglycine (DMG); (19) methylguanidine; (20) trimethylamine (TMA); (21) creatinine; (22) trimethylamine-*N*-oxide (TMAO); (23) choline, (24) glycine; (25) glycerol; (26) creatine; (27) α-glucose; (28) β-glucose; (29) betaine; (30) hippurate; (31) allantoin; (32) fumarate; (33) *N*(1)-methyl-4-pyridone-5-carboxamide; (34) 3-hydroxyphenyl propionic acid; (35) tryptophan; (36) phenylalanine; (37) phenylacetylglycine; (38) indoxyl sulfate; (39) uridine; (40) hypoxanthine; (41) formate. The numbers “×8” and “×2” stand for the amplification factor of spectra.

**Figure 4 ijms-20-03655-f004:**
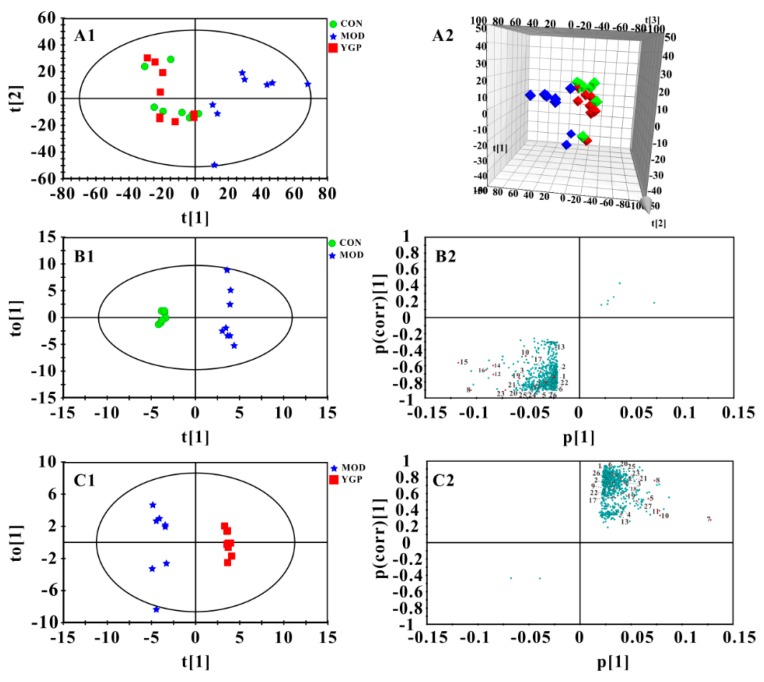
Results of multivariate statistical analysis. Principle component analysis (PCA) score plot and 3D-PCA score plot among CON, MOD, and YGP groups (**A1**,**A2**, *R*^2^*X* = 0.856, *Q*^2^ = 0.520). Orthogonal partial least squares discriminant analysis (OPLS-DA) score plot and *S*-plot between CON and MOD groups (**B1**,**B2**, *R*^2^*X* = 0.842, *R*^2^*Y* = 0.993, *Q*^2^ = 0.840, *p*-value of CV-ANOVA = 0.0298); OPLS-DA score plot and *S*-plot between MOD and YGP groups (**C1**,**C2**, *R*^2^*X* = 0.683, *R*^2^*Y* = 0.987, *Q*^2^ = 0.714, *p*-value of CV-ANOVA = 0.0370). The greater the distance of the red dots in *S*-plots from the origin, the greater the contribution to the grouping. Variables labeled with red dots could be considered as differential metabolites (**B2**,**C2**) and the labels next to red dots correspond to the numbers of metabolites in Appendix A. All the points of the important variable values (VIP) < 1 in the *S*-plots were removed, leaving only the variables with VIP > 1.

**Figure 5 ijms-20-03655-f005:**
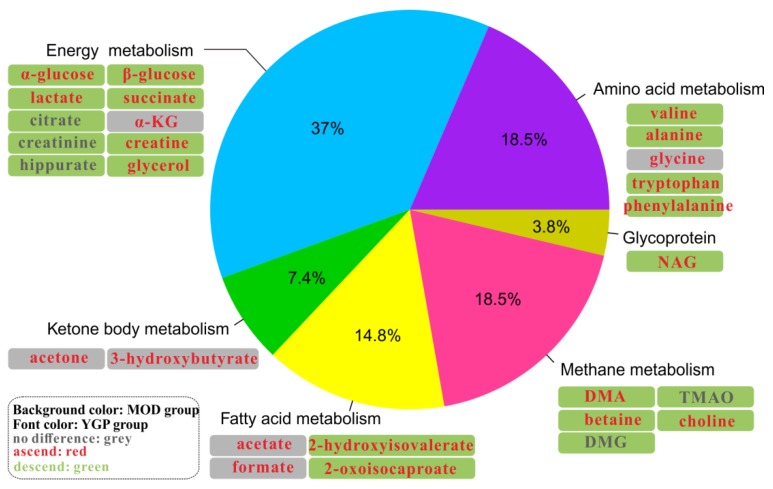
The distribution of 27 differential metabolites. The background color represents the MOD group vs. CON group, and color of font represents the YGP group vs. MOD group. Red means ascending, green means descending, and grey meansno significant change.

**Figure 6 ijms-20-03655-f006:**
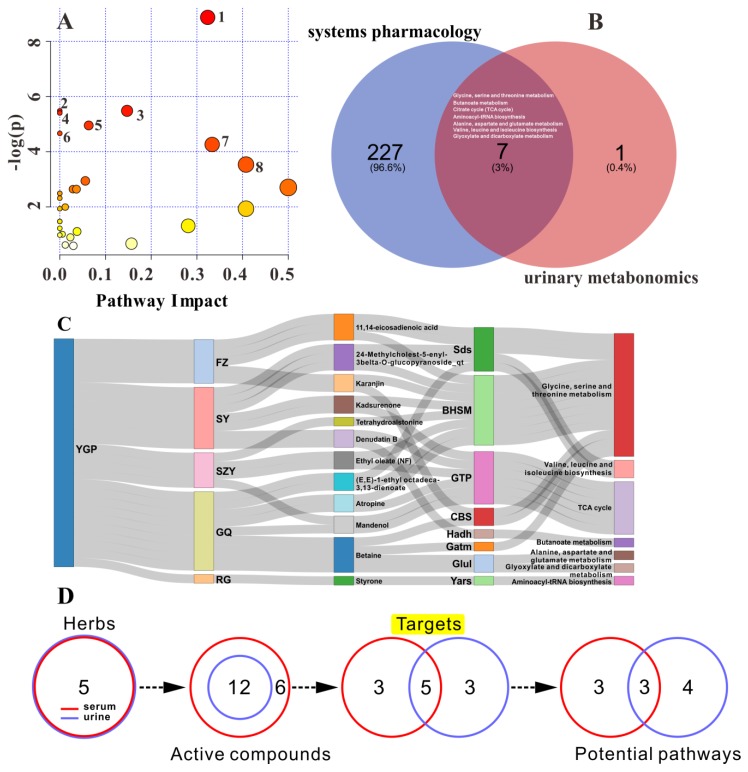
(**A**) The metabolic pathways corresponding to differential metabolites in urine sample; and (**B**) overlapping pathways between systems pharmacology and urinary metabonomics analysis. The numbers in **A** correspond to the serial numbers of the metabolic pathway in Appendix A, while the white pathway in **B** represents seven overlapping metabolic pathways. (**C**) The network of YGP–herbs–active compounds–target proteins–pathways. (**D**) Comparison of metabonomics and systems pharmacology studies between urine and serum samples.

**Figure 7 ijms-20-03655-f007:**
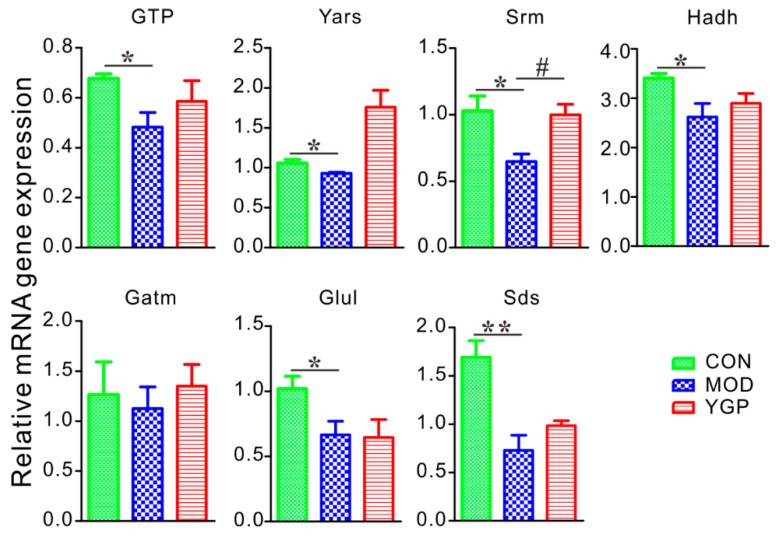
YGP regulates KYDS-related differential gene expression. All genes expression are examined by RT-qPCR and normalized to β-actin expression. Data represent mean ± SEM for at least three independent experiments; * *p* < 0.05, ** *p* < 0.01; ^#^
*p* < 0.05, ^##^
*p* < 0.01.

**Figure 8 ijms-20-03655-f008:**
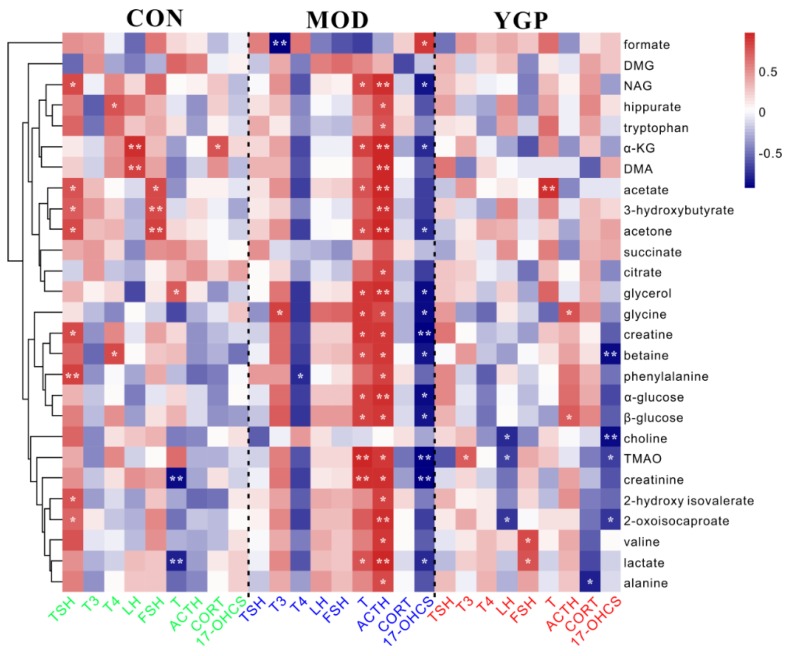
Pearson’s correlation analysis of 27 differential metabolites and nine biochemical parameters related to the pathogenesis axis of KYDS in the CON, MOD, and YGP groups, respectively. * *p* < 0.05, ** *p* < 0.01 (two-sided test), red indicates positive correlation, blue represents a negative correlation. 27 differential metabolites were clustered analysis by Euclidean distance.

**Figure 9 ijms-20-03655-f009:**
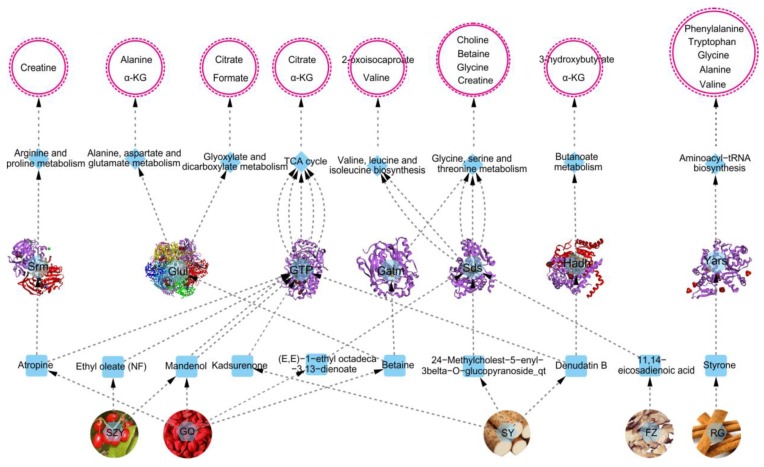
The overall interactive network diagram (herbs, active compounds, target proteins, pathways, and metabolites). SZY: *Fructus Corni*; GQ: *Fructus Lycii*; SY: *Rhizoma Dioscoreae*; FZ: *Radix Aconiti Lateralis Prreparata*; RG: *Cinnamomi cortex*. GTP: phosphoenolpyruvate carboxykinase; Yars: tyrosyl-tRNA synthetase; Srm: spermidine synthase; Sds: L-serine dehydratase; Hadh: 3-hydroxyacyl-CoA dehydrogenase; Glul: glutamine synthetase; Gatm: glycine amidinotransferase.

**Table 1 ijms-20-03655-t001:** Changes in biochemical parameters related to the pathogenesis axis of kidney-yang deficiency syndrome (KYDS) rats.

	Thyroid Gland			Testis			Adrenal Gland		
	TSH (UIU/mL)	T_3_ (nmol/L)	T_4_ (nmol/L)	LH (MIu/mL)	FSH (mIU/mL)	T (ng/dL)	ACTH (ng/mL)	CORT (ng/mL)	17-OHCS (nmol/mL)
CON	5.58 ± 0.41	6.11 ± 0.47	84.05 ± 2.81	3.16 ± 0.13	4.07 ± 0.160	178.71 ± 1.93	44.76 ± 3.33	150.67 ± 2.95	4.25 ± 0.46
MOD	4.30 ± 0.31 **	4.33 ± 049 **	44.58 ± 3.40 **	1.46 ± 0.31 **	2.32 ± 0.31 **	21.85 ± 1.44 **	28.95 ± 1.90 **	65.53 ± 4.63 **	3.69 ± 0.37 *
YGP	5.12 ± 0.46 ^##^	5.51 ± 0.45 ^##^	62.87 ± 2.54 ^##^	2.64 ± 0.08 ^##^	3.71 ± 0.66 ^##^	138.26 ± 6.54 ^##^	37.47 ± 1.56 ^##^	120.14 ± 4.38 ^##^	4.29 ± 0.57 ^#^

Note: values were presented as mean ± SD (*n* = 8). * As compared with control group (CON), * *p* < 0.05, ** *p* < 0.01, ^#^ as compared with KYDS group (MOD), ^#^
*p* < 0.05, ^# #^
*p* < 0.01.

**Table 2 ijms-20-03655-t002:** Primer sequence for quantitative real-time PCR analysis (RT-qPCR).

No	Gene	Primer Sequence
1	Phosphoenolpyruvate carboxykinase (GTP)	Forward: 5′-CTGCATAACGGTCTGGACTTC-3′
		Reverse: 5′-CAGCAACTGCCCGTACTCC-3′
2	Tyrosyl-tRNA synthetase (Yars)	Forward: 5′-GCTGCATCTTATCACCCGGAA-3′
		Reverse: 5′-GATCTTGGACATGGGTACAAAGT-3′
3	Spermidine synthase (Srm)	Forward: 5′-ACATCCTCGTCTTCCGCAGTA-3′
		Reverse: 5′-GGCAGGTTGGCGATCATCT-3′
4	L-serine dehydratase (Sds)	Forward: 5′-GAAGACCCCACTTCGTGACAG-3′
		Reverse: 5′-TCTTGCAGAGATGCCCAATGC-3′
5	3-hydroxyacyl-CoA dehydrogenase (Hadh)	Forward: 5′-TCAAGCATGTGACCGTCATCG-3′
		Reverse: 5′-TGGATTTTGCCAGGATGTCTTC-3′
6	Glutamine synthetase (Glul)	Forward: 5′-TGAACAAAGGCATCAAGCAAATG-3′
		Reverse: 5′-CAGTCCAGGGTACGGGTCTT-3′
7	Glycine amidinotransferase (Gatm)	Forward: 5′-GCTTCCTCCCGAAATTCCTGT-3′
		Reverse: 5′-CCTCTAAAGGGTCCCATTCGT-3′
8	Alcohol dehydrogenase 1C (Alcohol-1C)	Forward: 5′-ATGGGCACCGCTGGAAAAG-3′
		Reverse: 5′-TAACACGGACTTCCTTAGCCT-3′
9	Alcohol dehydrogenase class-3 (Alcohol-3C)	Forward: 5′-AGTTCGGATTAAGATCCTTGCCA-3′
		Reverse: 5′-ACTTTCCACAATTCCAGCACC-3′
10	Betaine-homocysteine S-methyltransferase 1 (BHSM)	Forward: 5′-TGTATGGGCTGTCGAAGTCTT-3′
		Reverse: 5′-CATGGTCTGCAAGCTAGTCCA-3′
11	Cystathionine beta-synthase (CBS)	Forward: 5′-CCAGGCACCTGTGGTCAAC-3′
		Reverse: 5′-GGTCTCGTGATTGGATCTGCT-3′
12	β-actins	Forward: 5′-GGAGATTACTGCCCTGGCTCCTA-3′
		Reverse: 5′-GACTCATCGTACTCCTGCTTGCTG-3′

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
