# Peer review of "Integrated Systems Pharmacology, Urinary Metabonomics, and Quantitative Real-Time PCR Analysis to Uncover Targets and Metabolic Pathways of the You-Gui Pill in Treating Kidney-Yang Deficiency Syndrome"

_ijms, 2019, doi:10.3390/ijms20153655_

Round 1
Reviewer 1 Report
I read with interest the manuscript entitled " Integrated Systems Pharmacology, Urinary
Metabonomics, and Quantitative Real-time PCR Analyses to Uncover Targets and Metabolic Pathways of You-gui Pill in Treating Kidney-yang Deficiency Syndrome, by Ruiqun Chen, et al” that is intended to be published in International Journal Molecular Sciences.
The study is well planned, performed, explained and discussed. In some parts is a very exhaustive manuscript. No concerns with its publication in the present form
Only few corrections in the English grammar
Author Response
Dear Editor/Reviewer:
We are truly grateful to yours’ critical comments and thoughtful suggestions on our manuscript (Integrated Systems Pharmacology, Urinary Metabonomics, and Quantitative Real-time PCR Analysis to Uncover Targets and Metabolic Pathways of You-gui Pill in Treating Kidney-yang Deficiency Syndrome). Based on these comments and suggestions, we have made careful modifications on the original manuscript, and uploaded it to the webpage. All changes made to the text are clearly highlighted in red color. Furthermore, we have consulted native English speakers for paper revision before the submission. We hope the revised manuscript will meet your Journal’s standard. Below you will find our point-by-point responses to the reviewers’ comments/questions.
We hope that these revisions are satisfactory and that the revised file will be acceptable for publication in International Journal of Molecular Sciences.
Thank you very much for your work concerning our paper.
Wish you all the best!
Yours sincerely,
Ruiqun Chen, Jia Wang, Runhua Zhan, Lei Zhang and Xiufeng Wang
Response to Reviewer 1 Comments
Comments and Suggestions for Authors
I read with interest the manuscript entitled “Integrated Systems Pharmacology, Urinary Metabonomics, and Quantitative Real-time PCR Analyses to Uncover Targets and Metabolic Pathways of You-gui Pill in Treating Kidney-yang Deficiency Syndrome, by Ruiqun Chen, et al” that is intended to be published in International Journal Molecular Sciences.
Point 1: The study is well planned, performed, explained and discussed. In some parts is a very exhaustive manuscript. No concerns with its publication in the present form.
Response 1: Thanks you for your nice comments on our manuscript.
Point 2: Only few corrections in the English grammar.
Response 2: Thanks for your careful checks. As you are concerned, there are several corrections in the English grammar that need to be addressed. We feel sorry for our poor writings. We have consulted native English speakers to help polish our article, and thoroughly inspect and modify the entire manuscript carefully. We hope the revised manuscript could be acceptable for you.

Reviewer 2 Report
The authors investigated the pharmacological mechanisms of You Gui Pill (YGP) for the treatment of Kidney Yang Deficiency (KYD) syndrome. My comments and concerns are as follows:
The KYD syndrome is not a well-defined syndrome and in human can be associated with different pathogenic mechanisms and clinical manifestations.
It remains questionable whether the animal model proposed by the authors can simulate KYD syndrome in human.
Author Response
Dear Editor/Reviewer:
We are truly grateful to yours’ critical comments and thoughtful suggestions on our manuscript (Integrated Systems Pharmacology, Urinary Metabonomics, and Quantitative Real-time PCR Analysis to Uncover Targets and Metabolic Pathways of You-gui Pill in Treating Kidney-yang Deficiency Syndrome). Based on these comments and suggestions, we have made careful modifications on the original manuscript, and uploaded it to the webpage. All changes made to the text are clearly highlighted in red color. Furthermore, we have consulted native English speakers for paper revision before the submission. We hope the revised manuscript will meet your Journal’s standard. Below you will find our point-by-point responses to the reviewers’ comments/questions.
We hope that these revisions are satisfactory and that the revised file will be acceptable for publication in International Journal of Molecular Sciences.
Thank you very much for your work concerning our paper.
Wish you all the best!
Yours sincerely,
Ruiqun Chen, Jia Wang, Runhua Zhan, Lei Zhang and Xiufeng Wang
Response to Reviewer 2 Comments
Comments and Suggestions for Authors
The authors investigated the pharmacological mechanisms of You Gui Pill (YGP) for the treatment of Kidney Yang Deficiency (KYD) syndrome. My comments and concerns are as follows:
Point 1: The KYD syndrome is not a well-defined syndrome and in human can be associated with different pathogenic mechanisms and clinical manifestations.
Response 1: Thank you very much, and this is a very enlightening question.
As you mentioned, The KYDS is not a well-defined syndrome, and our experiment is only based on animal models (rats) to study the pathogenesis of KYDS and the regulating effects of YGP on KYDS. Hence, there may be different pathogenic mechanisms and clinical manifestations in humans. We believe that this part of research is very necessary and is also a key step to clinical transformation. Therefore, on the basis of this study, we will conduct an in-depth study combined with samples of clinical patients in the next step, so as to verify the conclusions of this study.
We have supplemented relevant explanation in the manuscript:
“Given some of herbs, active compounds, targets, metabolic pathways, and differential metabolites that YGP acts on KYDS were found in this study by integrated systems pharmacology, urinary metabonomics, and RT-qPCR analysis, more subsequent experiments and assay, such as samples of clinical patients, western blot, targeted metabonomics and molecular pharmacology, are needed to verify the relevant results from different levels.”
(See line 537 at the last paragraph of 3.5. Fatty Acid Metabolism.)
Point 2: It remains questionable whether the animal model proposed by the authors can simulate KYD syndrome in human.
Response 2: We feel great thanks for your professional review work on our manuscript.
Firstly, the use of hydrocortisone to inject the leg muscles of rats/mouse to induce KYDS is the earliest established animal model of TCM syndrome, and is also the method used by most researchers to study KYDS [1-5]. Secondly, KYDS patients with clinically are mainly diagnosed by combining the clinical symptoms and 17-OHCS indicator [6-9]. Because a large number of studies have shown that 17-OHCS was significantly decreased when KYDS occurs, and it can be used as a biomarker of KYDS. Meanwhile, our study also integrates clinical symptoms (e.g. weight and behavioral change), and multiple biochemical indicators (TSH, T3, T4, LH, FSH, T, ACTH, and CORT) of target gland axis to define KYDS. Finally, we perform histopathological analysis of multiple tissues (hypothalamus, pituitary, thyroid, adrenal gland, and testis, etc) to define KYDS. All these indicators and pathological analysis have also been reported in many studies on KYDS [7, 8, 10-13]. In summary, we believe that KYDS rat model established in this study can basically simulate KYD syndrome in human. However, KYDS should also be further analyzed ultimately in combination with clinical patients, so as to truly reveal the mechanism of occurrence and development of KYDS.
Overall, thank you for your wonderful comments again. Your comments and suggestions have contributed a lot to improve the quality of our manuscript and subsequent research. If there are still any questions in our manuscript, please let us know, we will carefully modify and discuss it in the manuscript according to your opinion.
References
1. Tong, J. F.; Xu, Z. W.; Yang, Y. X.; Chen, Y. T.; Chen, H. S.; Zhou, X. J.; Li, C. Y. Comparative study on syndrome of deficiency of kidney yang rat model induced by adenine and hydrocortisone. China Journal of Traditional Chinese Medicine & Pharmacy 2015, 30, 3901-3904.
2. Zou, Z. J.; Gong, M. J.; Xie, Y. Y.; Wang, S. M.; Liang, S. W. Urinary metabonomic study of kidney-yang deficiency syndrome induced by hydrocortisone. Chinese Journal of Experimental Traditional Medical Formulae 2012, 18, 133-136.
3. Jiang, H.; Wang, Y. Z.; Hu, J. R. Influence of Jingang Wine on Rat Model and Mouse Model with Deficiency of Kidney Yang. Chinese Journal of Experimental Traditional Medical Formulae 2011, 17, 211-214.
4. Du, J.; Li, N. Methods for preparing kidney deficient models and related evaluating factors. Journal of Clinical Rehabilitative Tissue Engineering Research 2010, 14, 9433-9466.
5. Chen, M.; Zhao, L.; Jia, W. Metabonomic study on the biochemical profiles of a hydrocortisone-induced animal model. J. Proteome Res. 2005, 4, 2391-2396.
6. Zhao, L.; Wu, H.; Qiu, M.; Sun, W.; Wei, R.; Zheng, X.; Yang, Y.; Xin, X.; Zou, H.; Chen, T.; Liu, J.; Lu, L.; Su, J.; Ma, C.; Zhao, A.; Jia, W. Metabolic signatures of kidney yang deficiency syndrome and protective effects of two herbal extracts in rats using GC/TOF MS. Evid. Based Complement. Alternat. Med. 2013, 2013, 540957.
7. Liu, Q.; Zhang, A.; Wang, L.; Yan, G.; Zhao, H.; Sun, H.; Zou, S.; Han, J.; Ma, C. W.; Kong, L.; Zhou, X.; Nan, Y.; Wang, X. High-throughput chinmedomics-based prediction of effective components and targets from herbal medicine AS1350. Sci. Rep. 2016, 6, 38437.
8. Zhang, A.; Liu, Q.; Zhao, H.; Zhou, X.; Sun, H.; Nan, Y.; Zou, S.; Ma, C. W.; Wang, X. Phenotypic characterization of nanshi oral liquid alters metabolic signatures during disease prevention. Sci. Rep. 2016, 6, 19333.
9. Chen, X.; Hu, C.; Dai, J.; Chen, L. Metabolomics analysis of seminal plasma in infertile males with kidney-yang deficiency: a preliminary study. Evid. Based Complement. Alternat. Med. 2015, 2015, 1-8.
10. Zhou, X. H.; Zhang, A. H.; Wang, L.; Tan, Y. L.; Guan, Y.; Han, Y.; Sun, H.; Wang, X. J. Novel chinmedomics strategy for discovering effective constituents from ShenQiWan acting on ShenYangXu syndrome. Chinese journal of natural medicines 2016, 14, 561-581.
11. Nan, Y.; Zhou, X.; Liu, Q.; Zhang, A.; Guan, Y.; Lin, S.; Kong, L.; Han, Y.; Wang, X. Serum metabolomics strategy for understanding pharmacological effects of ShenQi pill acting on kidney yang deficiency syndrome. J. Chromatogr. B Analyt. Technol. Biomed. Life Sci. 2016, 1026, 217-226.
12. Li, B.; Luo, Q. L.; Nurahmat, M.; Jin, H. L.; Du, Y. J.; Wu, X.; Lv, Y. B.; Sun, J.; Abduwaki, M.; Gong, W. Y.; Dong, J. C. Establishment and comparison of combining disease and syndrome model of asthma with "kidney yang deficiency" and "abnormal savda". Evid. Based Complement. Alternat. Med. 2013, 2013, 658364.
13. Zhang, L.; Han, X.; Li, Z.; Liu, R.; Xu, W.; Tang, C.; Wang, X.; Xiao, H. Metabolomics research on time-selected combination of Liuwei Dihuang and Jinkui Shenqi pills in treating kidney deficiency and aging by chemometric methods. Chemometrics Intellig. Lab. Syst. 2014, 130, 50-57.
